# Application of a New Polymer Particle Adhesive for Lithium Battery Separators

Guanghua Huang [1,2], Haohan Wu [1], Gongxun Cao [1], Zhijun Liu [1], Hanlin Hu [1,*] and Shifeng Guo [2,3,*]

1. Hoffmann Institute of Advanced Materials, Shenzhen Polytechnic, 7098 Liuxian Blvd, Shenzhen 518055, China
2. Shenzhen Key Laboratory of Smart Sensing and Intelligent Systems, Shenzhen Institute of Advanced Technology, Chinese Academy of Sciences, 1068 Xueyuan Avenue, Shenzhen 518055, China
3. Guangdong Provincial Key Laboratory of Robotics and Intelligent System, Shenzhen Institute of Advanced Technology, Chinese Academy of Sciences, 1068 Xueyuan Avenue, Shenzhen 518055, China
* Correspondence: hanlinhu@szpt.edu.cn (H.H.); sf.guo@siat.ac.cn (S.G.)

**Abstract:** Lithium battery separators play a critical role in the performance and safety of lithium batteries. In this work, four kinds of polymer particle adhesives (G1–G4) for lithium battery separators were synthesized via dispersion polymerization, using styrene, butyl acrylate and acrylonitrile as monomers. The particle size/size distributions, particle morphologies and glass transition temperatures (Tg) of polymer particle adhesives were explored using laser particle size analysis, scanning electron microscopy (SEM) and differential scanning calorimetry (DSC), respectively. The adhesion strengths between the battery separators and the poles piece were examined using a tensile machine. The prepared polymer particle adhesive with a uniform distribution of particle size was obtained when the mass ratio of ethanol to water reached 85:15. Compared with the other three polymer particle adhesives, the prepared G3 coated on the surface of the battery separator exhibited a stronger adhesion with the battery pole piece. In addition, the Land battery test system was applied to examine the electrochemical performance of the lithium battery assembled with the battery separator with the prepared polymer particle adhesives. The results suggest that the electrochemical performance of the lithium battery assembled with the battery separator with polymer particle adhesive G3 is the best among the four counterparts.

**Keywords:** polymer particles; adhesion force; battery separator; lithium battery

## 1. Introduction

Recently, with the rapid development of the economy, the subsequent environmental pollution problems have become more and more prominent, and the view of carbon neutrality has constantly been a cause for concern around the world. Therefore, the need to vigorously develop electric vehicles has become the consensus around the world [1–3]. The development of electric vehicles has also become the focus of many large enterprises [4]. The safety performance of electric vehicles has attracted the attention of consumers, especially the lithium batteries used in electric vehicles.

Lithium batteries are mainly composed of positive and negative electrode sheets, a battery separator and a battery electrolyte. In lithium batteries, the safety performance of the battery separator affects the performance of the lithium battery [5,6]. As a key part placed between the positive and negative electrode sheets of the lithium battery, the battery separator is mainly used as a carrier for effective transmission of lithium ions between the positive and negative electrode sheets. It can also prevent short circuit caused by direct contact between the positive and negative electrode sheets [7–9]. Therefore, the battery separator must have chemical/electrochemical stability and sufficient mechanical strength to avoid damage during the assembly of the lithium battery [10–14]. The battery separator used in lithium batteries is made of polyolefin resin, including polyethylene (PE), polypropylene (PP) and their blends, which can be obtained by dry and wet processes [15–19].

However, the polyolefin separator sold in the market has a series of problems, and it is difficult to use directly in lithium batteries. The polyolefin separator is made of an inert material, which is not well wetted by organic electrolyte solutions in lithium batteries. Additionally, it has a poor ability to hold the electrolyte during charge–discharge cycles of the lithium battery. The polyolefin separator is also a flexible material that cannot meet the requirements of a lithium battery for the thermal stability of the battery separator. Therefore, the polyolefin separator must be modified to meet the requirements of lithium batteries. Costa et al. dispersed aluminum trioxide ($Al_2O_3$) nanoparticles and polyvinylidene fluoride (PVDF) in deionized water to prepare an aqueous suspension containing $Al_2O_3$ and PVDF, which was coated on a polyolefin separator surface. The mechanical properties and puncture resistance of the polyolefin separator were enhanced, and the heat resistance improved. This can also cause the polyolefin separator to have an affinity with the battery electrolyte after modification, thereby improving the safety and cycling performance of the lithium battery [20]. Previous studies reported [21] that polyimide (PI), polyetherimide (PEI) or polyacrylonitrile (PAN) was coated on the surface of the polyolefin separator to enhance its affinity to the battery electrolyte. Zhu et al. grafted vinyltrimethoxysilane (VTMS) on the surface of PE film with electron beam (EB) irradiation. The VTMS particles were hydrolyzed and silica was introduced, and, thus, the thermal stability of the PE film was improved [22].

On the one hand, in the lithium battery manufacturing industry, surface coating of polyolefin separators has become more convenient with the widespread use of large gravure coaters. Without affecting the air permeability of the polyolefin separator, coating $Al_2O_3$ nanoparticles with a certain thickness on the surface of the polyolefin separator is beneficial to improve the thermal resistance of the polyolefin separator, reduce its thermal contractility and increase its affinity for the battery electrolyte. This method is widely used in the preparation of lithium battery separators. On the other hand, during the preparation of the lithium battery, the position between the battery separator and pole piece may be shifted when the adhesion between them is weak. This causes a short circuit with direct contact between the positive and negative pole of the lithium battery. Thus, polyvinylidene fluoride (PVDF) is also used as the adhesive between battery separators and pole pieces in most lithium battery manufacturers. Although the adhesive has excellent electrochemical performance, it has the shortcomings of insufficient adhesion and a complex coating process. Moreover, the synthesis process of PVDF that is used in lithium batteries is complex, so its price has always been high. In view of the problems and defects of PVDF, some companies, such as Indigo in China and Roen in Japan, have developed some polyacrylate adhesives for lithium battery separators. The electrochemical performance of lithium batteries is good. However, the synthesis process of the polymer particle adhesive is complicated and its price is high. Therefore, it is particularly important to develop a kind of polymer particle adhesive that can be used for lithium battery separators.

Hence, we have synthesized four kinds of polymer particle adhesives that can be coated on the surface of battery separators. They can generate adhesion between the battery separator and the pole piece, which is measured by a tensile machine. Additionally, the particle size distributions, particle morphologies and glass transition temperatures of polymer particle adhesives were investigated. The battery separator coated with polymer particle adhesive was used to assemble lithium batteries, and the electrochemical properties of the lithium batteries were tested and analyzed by Land battery test system, mainly for the CV, EIS and charge–discharge cycle of the lithium batteries.

## 2. Experimental

### 2.1. Materials

Solvents and reagents were purchased and used without any purification unless noted otherwise. These were styrene (St, purity: 99%), n-butylacrylate (*n*-BA, purity: 99%), acrylonitrile (ACN, purify: 99%), N-methyl pyrrolidone (NMP, purify: 99%), azodiisobu-tyronitrile (AIBN, 99%) divinylbenzene (DVB, purify: 80%) and polyvinyl pyrrolidone

(PVPk30) purchased from Shanghai Macklin Biochemical Co., Ltd., Shanghai, China. Ethyl alcohol absolute (EtOH, purity: 98%) purchased from Guangzhou Chemical Reagent Co., Guangzhou, China. Deionized water ($H_2O$) was made for laboratory. Lithium iron phosphate (LiFePO$_4$) purchased from Shenzhen Dynanonic Co., Shenzhen, China. Conductive carbon black (Super-P) and polyvinylidene fluoride (PVDF) purchased from Shenzhen Wedafr New Energy Technology Co., Shenzhen, China. The battery electrolyte containing lithium hexafluorophosphate (LiPF$_6$) and aluminum foil (AF) was purchased from Canrd Technology Co., Shenzhen, China. The battery separator was obtained from Shanghai Dinho Technology Co., Shanghai, China. A type of polymer particle adhesive (S-1, $d_{10}$ = 3.7 $\mu$m; $d_{50}$ = 5.1 $\mu$m; $d_{90}$ = 9.7 $\mu$m) was obtained from Si Chuan Indigo Co., Ltd., Chengdu, China.

*2.2. Synthesis of Polymer Particle Adhesive*

Figure 1 and Table 1 show the synthetic conditions of G1, G2, G3 and G4. For the synthesis of G1, G2 and G3, St (50 g, 0.48 mol), *n*-BA (45 g, 0.35 mol), ACN (5 g, 0.094 mol), DVB (1.5 g, 0.012 mol), AIBN (1 g, 0.006 mol) and PVPk30 (4.5 g) were added to a 1 L flask. The mixed solvents of EtOH (200 g) and $H_2O$ (200 g) for G1 or EtOH (280 g) and $H_2O$ (120 g) for G2 or EtOH (340 g) and $H_2O$ (60 g) for G3 were dripped into the reaction flask. The temperature of the reaction mixture was adjusted to 60 °C in nitrogen protection. The reactions were refluxed and condensed during the reaction. When the drop was finished, the reaction temperature was kept at 70 °C until the monomers were completely reacted. At the end of the reaction, the reaction solution in the flask was milky white, which was the desired product. Next, the prepared product was filtered and continuously cleaned with deionized water and EtOH (1:1) to remove impurities from the product. Finally, the filter residue was dispersed in deionized water to create the polymer particle adhesives used in this paper.

For the synthesis of G4, St (65 g, 0.624 mol), *n*-BA (30 g, 0.234 mol), ACN (5 g, 0.094 mol), DVB (1.5 g, 0.012 mol), AIBN (1 g, 0.006 mol) and PVPk30 (4.5 g) were added to a 1 L flask. The mixed solvents of EtOH (340 g) and $H_2O$ (60 g) were dripped into the reaction flask. The temperature of the reaction mixture was adjusted to 60 °C in nitrogen protection. The reactions were refluxed and condensed during the reaction. When the drop was finished, the reaction temperature was kept at 70 °C until the monomers were completely reacted. At the end of the reaction, the reaction solution in the flask was milky white, which was the desired product. Next, the prepared product was filtered and continuously cleaned with deionized water and EtOH (1:1) to remove impurities from the product. Finally, the filter residue was dispersed in deionized water to create the polymer particle adhesives used in this paper.

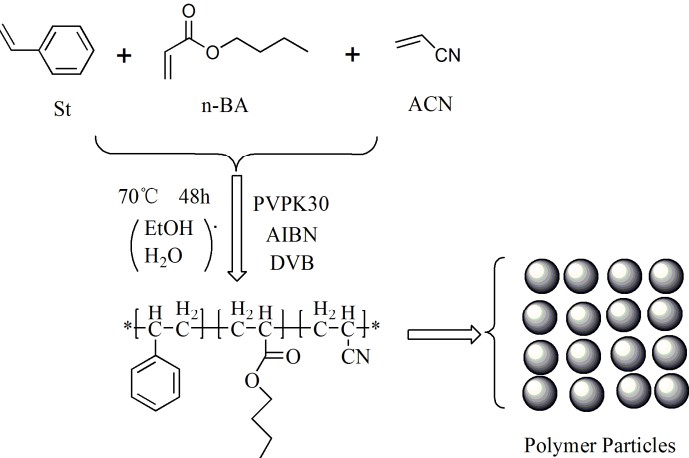

**Figure 1.** The synthesis process of the polymer particle adhesive.

**Table 1.** The synthetic formula of polymer particle adhesive.

| Number | m (St:$n$-BA:ACN) | m (EtOH:H$_2$O) | DVB | AIBN | PVPK30 | Temp | Time |
|---|---|---|---|---|---|---|---|
| G1 | 5:4.5:0.5 | 50:50 | 1.5% | 1% | 4.5% | 70 °C | 36 h |
| G2 | 5:4.5:0.5 | 70:30 | 1.5% | 1% | 4.5% | 70 °C | 36 h |
| G3 | 5:4.5:0.5 | 85:15 | 1.5% | 1% | 4.5% | 70 °C | 36 h |
| G4 | 6.5:3:0.5 | 85:15 | 1.5% | 1% | 4.5% | 70 °C | 24 h |

*2.3. Characterization of Physical and Chemical Properties of Materials*

The particle size distributions and the particle sizes ($d_{10}$, $d_{50}$, $d_{90}$) of the polymer particle adhesive samples were analyzed by a model BT-9300S laser diffraction particle size analyzer (Dandong Better Size Instrument Ltd., Dandong, China). A scanning electron microscope (ZEISS Instrument Ltd., Oberkochen, Germany) with a model of EVO-18 was used to observe the morphology and size of the polymer particle adhesive samples. The glass transition temperature (Tg) of polymer particle adhesive was characterized by a model Q500 thermogravimeter (TA Instruments Co., Framingham, USA) in N$_2$ at a heating rate of 20 °C/min to 200 °C. The coating of the polymer particle adhesive on the battery separator surface was achieved by using a wire rod coater (OSG Technology Co., Tokyo, Japan) of type OPS-1.5.

The adhesion value test process of the battery separator coated with the polymer particle adhesive is shown in the Figure 2. Firstly, scissors were used to cut the coated battery separator and battery pole piece into strips of 3 × 12 cm$^2$ and 3 × 10 cm$^2$, respectively. Then, they were neatly stacked together, and hot-pressed at 95 °C and 1 MPa for 5 min using a hot-pressing machine (Henchld Science Technology Co., Tianjin, China), model YPH-600C. Finally, the adhesion value between the battery separator and the battery pole piece was determined by a tensile machine (Jinan Xinzhun Instruments Co., Jinan, China), model XLW-H-500.

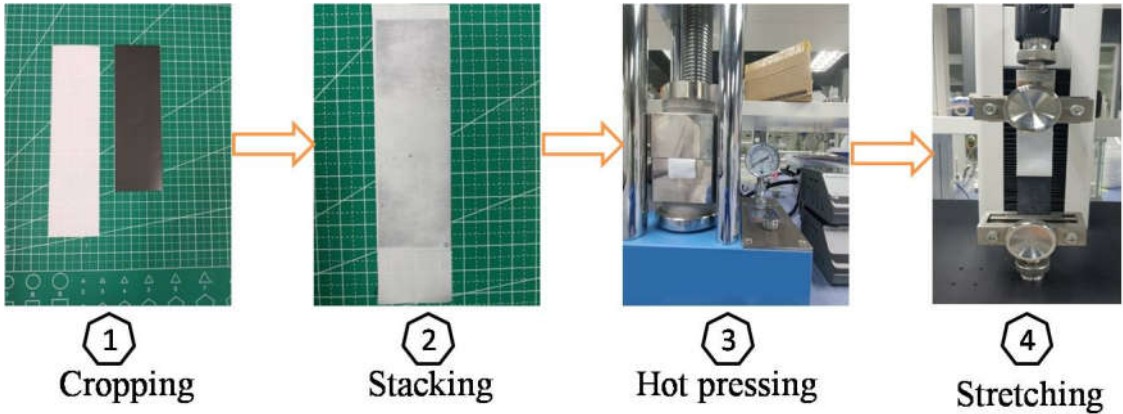

**Figure 2.** The adhesion test between battery separator and battery pole piece.

*2.4. Button Battery Assembly*

Preparation of positive electrode sheet: lithium iron phosphate powder (LiFeO$_4$), binder (PVDF) and conductive agent (Super-P) were weighed in a mass ratio of 8:1:1, and then placed in solvent NMP to stir and disperse for 12 h to obtain a positive electrode suspension. Then, the positive electrode suspension was coated on the aluminum foil (AF) with a coater (Shenzhen Kejingstar Technology Co., Shenzhen, China), model MSK-AFA-ES200, before drying it in an oven at 80 °C for 12 h, cutting it into 12 mm diameter discs with a microtome (Shenzhen Kejingstar Technology Co., Shenzhen, China) using model MSK-T10 (the 12 mm discs are the positive electrode sheets) and weighing them.

Preparation of battery separator: the battery separators coated with the different polymer particle adhesives (G3, G4 and S-1) were cut into 16 mm diameter discs with a microtome (Shenzhen Kejingstar Technology Co., Shenzhen, China) using model MSK-T10.

The button battery consists of a positive shell, a positive electrode sheet, a battery separator, a lithium sheet, a gasket, a leaf spring, a negative shell and a battery electrolyte. Among them, the battery shell is 2032 type, and the battery electrolyte is $LiPF_6$. The schematic diagram of button battery structure is shown in Figure 3. The main materials of the assembled lithium battery are listed in Table 2.

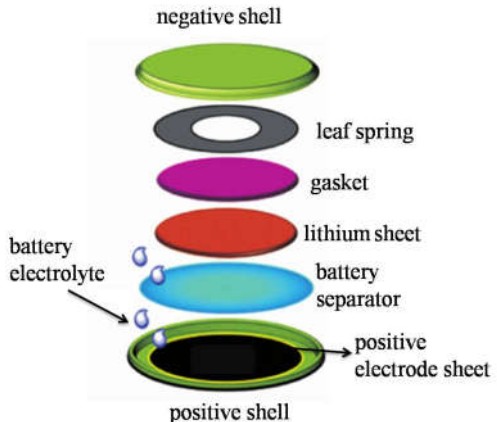

**Figure 3.** The sequence diagram of lithium battery assembly.

**Table 2.** The main materials of the assembled lithium battery.

| Type | Positive Electrode Sheet | Battery Electrolyte | Negative Electrode Sheet | Battery Separator |
|---|---|---|---|---|
| LG3 | $LiFePO_4$ | $LiPF_6$ | Lithium sheet | G3 coated |
| LG4 | $LiFePO_4$ | $LiPF_6$ | Lithium sheet | G4 coated |
| LS | $LiFePO_4$ | $LiPF_6$ | Lithium sheet | S-1 coated |

*2.5. Characterization of Electrochemical Properties*

(a)  Cyclic Voltammetry (CV)

Cyclic voltammetry (CV) is to perform multiple scan tests on the assembled lithium battery within the selected test range by controlling different scan voltages. During the scanning interval, the corresponding redox reaction occurs at the electrode, and the corresponding current–potential curve is recorded. In this paper, the test voltage range was 2.4–4.4 V, and the scanning speed was 0.1 mV/s.

(b)  Electrochemical Impedance Spectroscopy (EIS)

Electrochemical impedance spectroscopy is used to analyze the conductivity of electrode materials of assembled lithium batteries and the diffusion coefficient of lithium ions. From the electrochemical impedance spectroscopy, we can know the influence of the battery separator coated with a polymer particle adhesive on the assembled lithium battery, especially on the diffusion coefficient of lithium ion. In this paper, the high-frequency band was set as 100 KHz, and the low-frequency band was set as 0.01 Hz.

(c)  Charge and discharge performance test

The assembled lithium battery was charged and discharged with constant current through the Land battery test system (CT2001A), and the voltage window was 2.4–4.4 V. Under the constant temperature of 25 °C, the discharge capacity and capacity retention rate at different discharge rates was tested.

## 3. Results and Discussion

### 3.1. The Particle Size of the Synthesized Polymer Particle Adhesive

Figure 4 shows the particle size distributions and particle sizes ($d_{10}$, $d_{50}$ and $d_{90}$) of the synthesized polymer particle adhesives. Clearly, with the different ratios of ethanol and water in the mixed reaction solvents, there are large differences in the particle size distributions and particle sizes of the synthesized products. The particle size distributions of samples G1 and G2 are two peaks, which indicate that there are two sizes of polymer particles in these two samples. This is because the reaction solution is a heterogeneous system, in which an aqueous phase and an organic phase exist, when the mass ratio of ethanol and water is less than 7:3. Therefore, the reaction of monomers in the aqueous phase follows the emulsion polymerization mechanism to obtain nano-scale polymer particles. The reaction of monomers in the organic phase follows the dispersion polymerization mechanism to obtain micron-sized polymer particles [23]. There is only one peak in the particle size distributions of samples G3 and G4, which indicates that the particle sizes of these two samples are relatively uniform. The reason for this phenomenon is that the reaction solution is a homogeneous system when the mass ratio of ethanol and water is greater than 8:2. Therefore, the monomers in the reaction solution follow a dispersion polymerization mechanism to obtain micron-sized polymer particles [24].

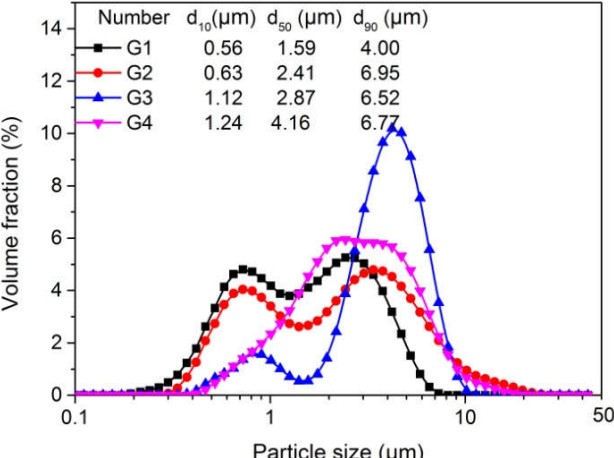

**Figure 4.** The particle size distribution and particle size ($d_{10}$, $d_{50}$ and $d_{90}$) of the synthesized polymer particle adhesive.

Figure 5 shows the SEM of the synthesized polymer particle adhesives (G1, G2, G3 and G4). Clearly, there are two sizes of polymer particles in samples G1 and G2, while the particle sizes of samples G3 and G4 are relatively uniform. The results of the SEM in Figure 5 are basically consistent with the results of the particle size distribution of synthesized polymer particle adhesives. Given that the particle sizes of polymer particle adhesives G1 and G2 were not uniform, these two adhesives were not used in battery separators in this paper.

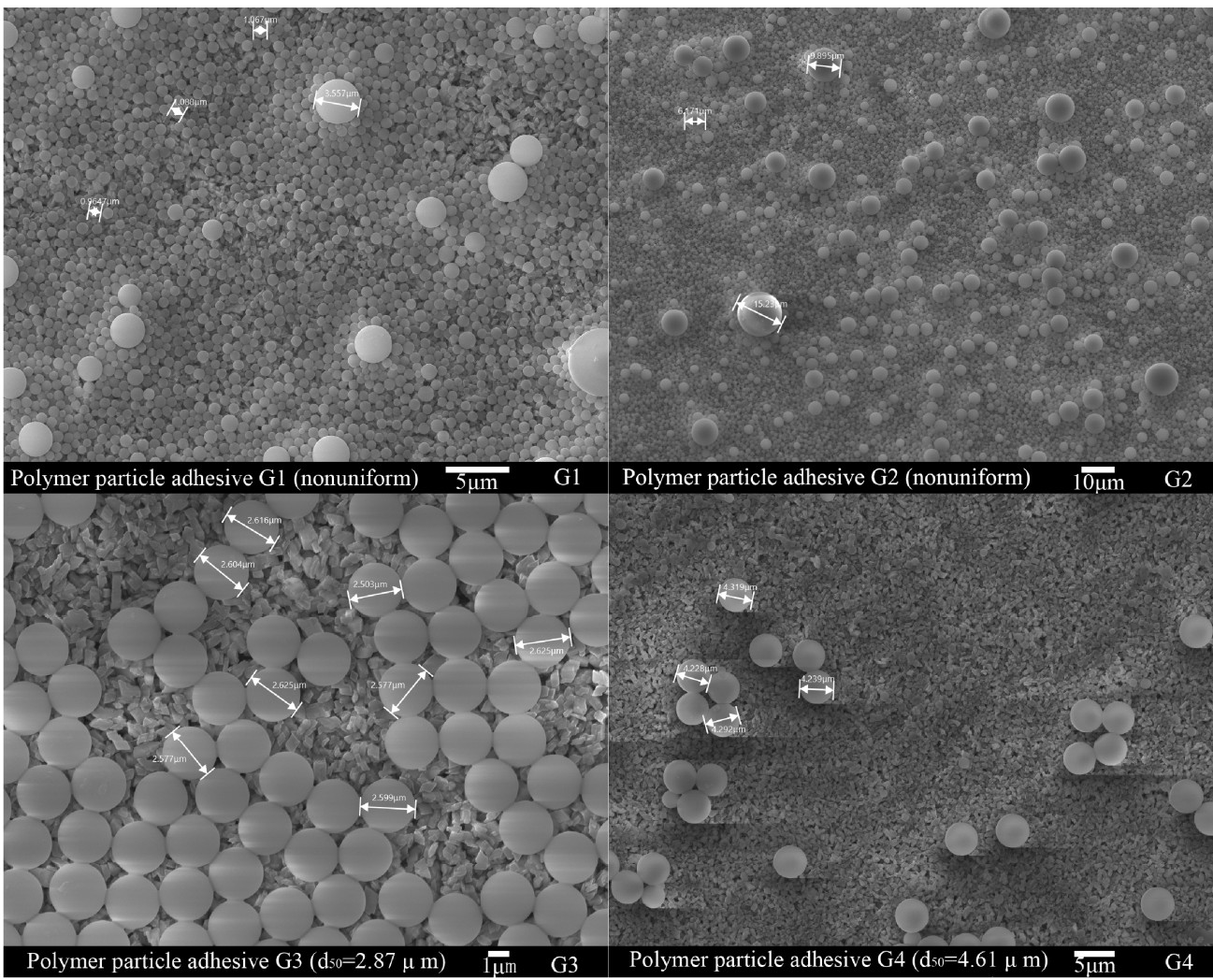

**Figure 5.** The SEM of the synthesized polymer particle adhesives (G1, G2, G3 and G4).

### 3.2. The Glass Transition Temperature

Figure 6 shows the DSC curves and glass transition temperatures (Tg) of the polymer particle adhesives. Clearly. The Tg values of samples G1, G2 and G3 are 33 °C because the monomer mass ratio of their synthesis reactions is St:*n*-BA:CAN = 5:4.5:0.5. The Tg value of sample G4 is 58 °C, which is significantly higher than that of samples G1, G2 and G3. This is mainly because the reactive monomers St and ACN are hard monomers, and the Tg values of polystyrene and polyacrylonitrile are 105 °C and 78 °C, respectively. The reaction monomer *n*-BA is a soft monomer, and the Tg value of polybutyl acrylate is −55 °C. The proportion of hard monomers in sample G4 is 70%, while the proportion of hard monomers in samples G1, G2 and G3 is 55%. Therefore, the Tg value of sample G4 is higher than that of samples G1, G2 and G3.

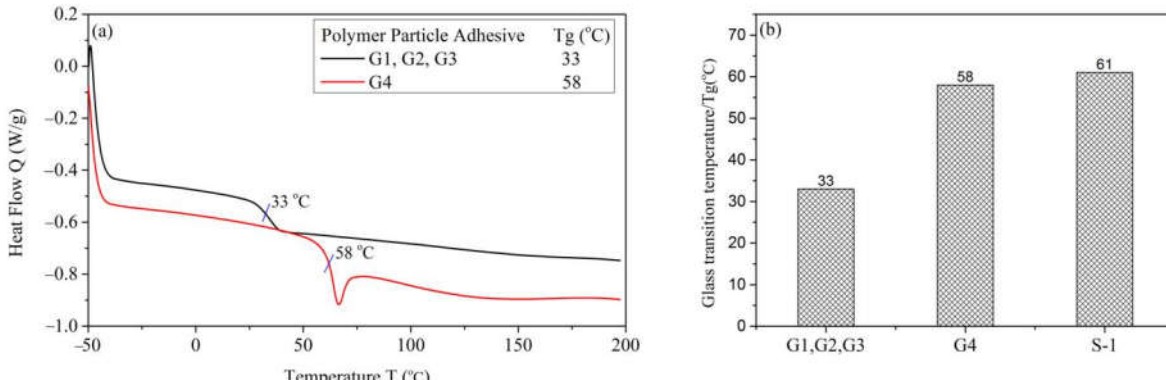

**Figure 6.** (**a**) The DSC curve of polymer particle adhesive (G1, G2, G3 and G4); (**b**) glass transition temperature (Tg) of polymer particle adhesive (G1, G2, G3, G4 and S-1).

### 3.3. The Coating of Polymer Particle Adhesive

Figure 7 shows the scanning electron microscope (SEM) images of polymer particles adhesives (G3, G4 and S-1) coated onto battery separator surface by wire rod coater. Clearly, the polymer particle adhesive coated on the battery separator is arranged in a single point rather than a stack. Table 3 shows that the coated areal densities of G3, G4 and S1 are 0.121 g/m$^2$, 0.183 g/m$^2$ and 0.22 g/m$^2$, respectively. For the lithium battery, the area density of the polymer particle adhesive on the battery separator is in the range of 0.5 g/m$^2$ to 0.25 g/m$^2$, which is in line with the use requirements of lithium batteries.

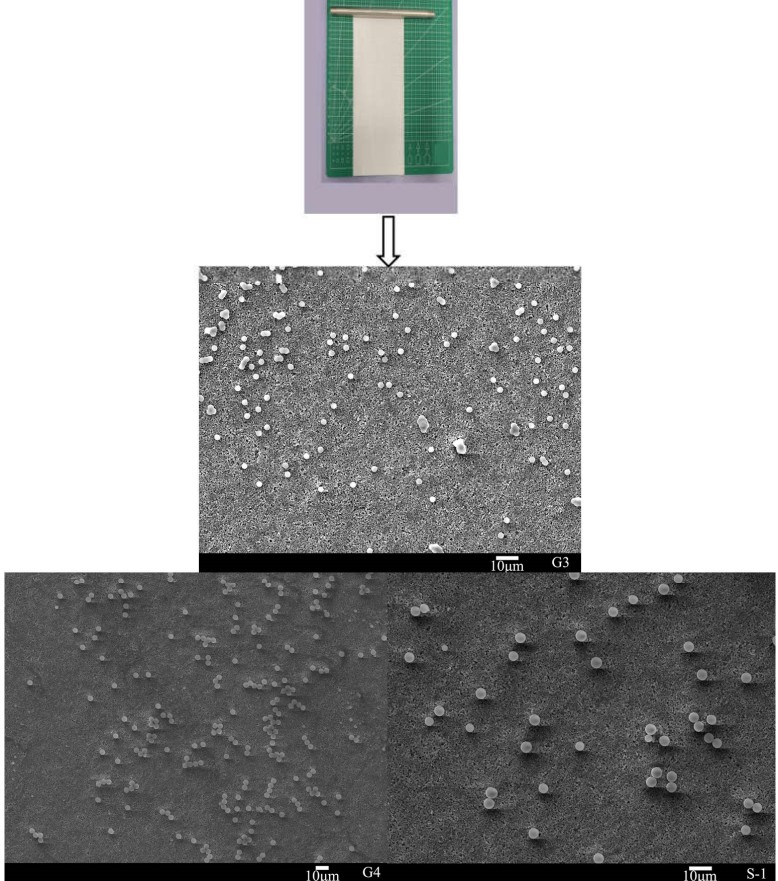

**Figure 7.** The SEM images of polymer particle adhesives (G3, G4 and S-1) coated onto the battery separator surface by wire rod coater.

**Table 3.** The coated areal density of battery separator surface after wire rod coating.

| Number | Tg (°C) | d$_{50}$ (μm) | Coated Areal Density (g/m$^2$) |
|--------|---------|---------------|--------------------------------|
| G3     | 33      | 2.87          | 0.121                          |
| G4     | 58      | 4.16          | 0.183                          |
| S-1    | 61      | 5.1           | 0.22                           |

*3.4. Adhesion Value between Battery Separator and Pole Piece*

The test method of the adhesion between the battery separator and the pole piece is shown in Figure 2, and the test results are shown in Figure 8a. Clearly, the adhesion between the battery separator and the positive electrode is greater than that between the battery separator and the negative electrode. This could be because the positive electrode material used in the battery is different from the negative electrode, which leads to the difference in the adhesion between them and the battery separator. In order to compare the adhesion between battery separators and pole pieces coated with different polymer particle adhesives, a normalization method is proposed in this paper. Figure 8b shows the adhesion value between the pole piece and the battery separator coated with different polymer particle adhesives under the unit areal density. It can be seen from Figure 8b that the adhesion value between the battery separator and the pole piece coated with sample G3 is the largest. This is because the Tg value of sample G3 is smaller than that of G4 and S-1, indicating that the content of soft monomer (*n*-BA) in sample G3 is more. The functional group that provides adhesion is mainly the ester group (-CO-O-) in the monomer *n*-BA [25]. In addition, the cyanide group (-CN) in the monomer ACN also provides part of the adhesion [26].

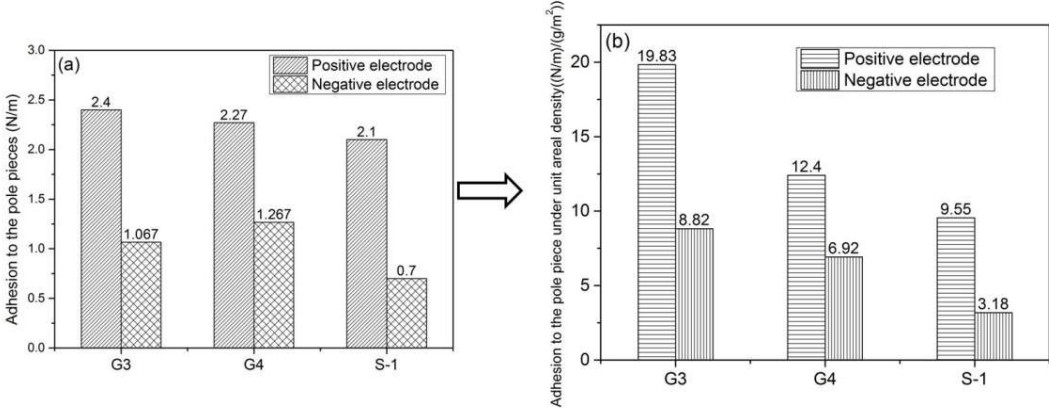

**Figure 8.** Adhesion values between battery separators and pole pieces. As the coating process cannot be precisely controlled, the results in (**a**) show the adhesion between the pole pieces and battery separators coated with polymer particle adhesive (G3, G4 and S-1) of different coated areal density (see Table 3). In order to solve the problem, the author divides the adhesion values in (**a**) by corresponding coating surface density of the battery separators, and obtains the adhesion values between the pole piece and the battery separator under the unit coating surface density of polymer particle adhesions, which is (**b**).

*3.5. Electrochemical Performance of Assembled Lithium Battery*

3.5.1. Cyclic Voltammetry Analysis

Figure 9 shows the cyclic voltammograms of lithium batteries assembled (LG3, LG4 and LS) with battery separators coated with G3, G4 and S-1. The lithium battery LS is a commercialized one, which is working as a control sample for the comparison with the ones we designed. The scanning speed was 0.1 mV/s and the scanning range was 2.2 V–4.4 V. Clearly, the lithium batteries prepared by the battery separators coated with

three kinds of polymer particle adhesive (G3, G4 and S-1) match the cyclic voltammetry curves of LiFePO$_4$. All of the lithium batteries show a pair of redox peaks in the scanning interval, corresponding to the charge/discharge process of iron phosphate/lithium iron phosphate [27–29]. It can be seen from Figure 9 that after the first cycle, the CV curves of the lithium battery have good coincidence and good cycle performance. In addition, it can be seen that the three kinds of lithium batteries have excellent reversibility through the peak shape of the redox peak. Therefore, it can be considered that the separators coated with different polymer binders have little effect on the cycle performance of the cathode material.

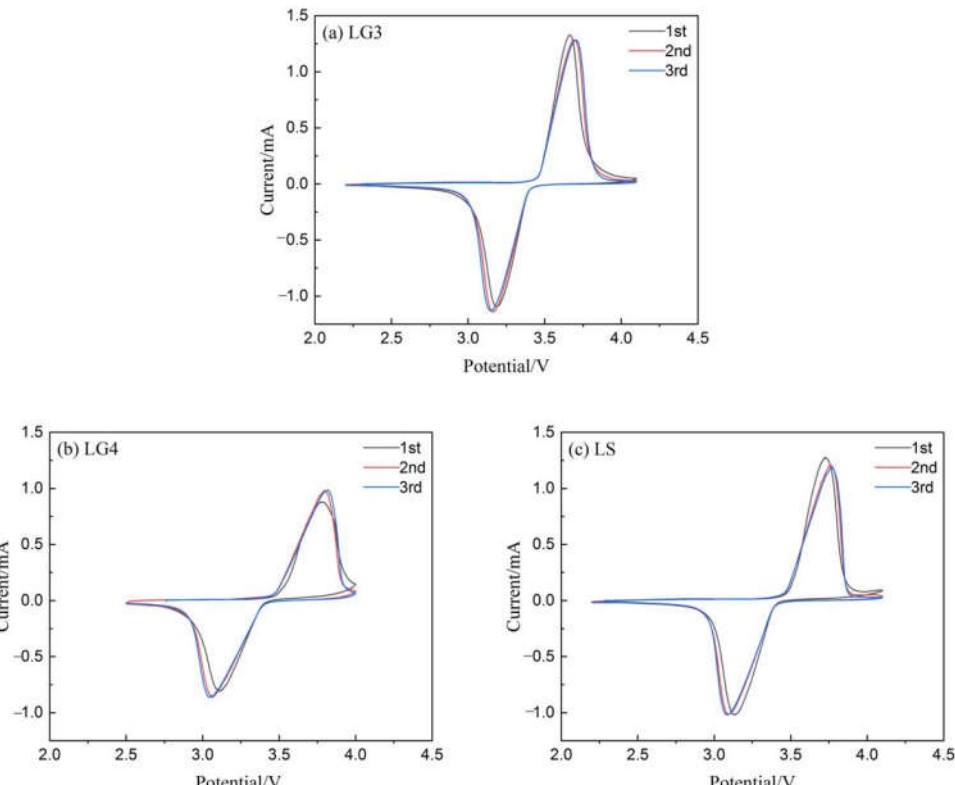

**Figure 9.** The cyclic voltammograms of the assembled lithium batteries.

Table 4 shows the data of peak voltage and potential difference in the cyclic voltam-mogram. The magnitude of the potential difference reflects the reversibility of the lithium battery. The potential difference of the lithium battery assembled by the battery separator coated with G3 is the smallest. This shows that the battery has the smallest polarization and the best reversible performance during the charging and discharging process.

**Table 4.** The data of peak voltage and potential difference in the cyclic voltammogram.

|  | 1st | | | 2nd | | | 3rd | | |
|---|---|---|---|---|---|---|---|---|---|
|  | Peak Voltage | | Potential Difference | Peak Voltage | | Potential Difference | Peak Voltage | | Potential Difference |
| LG3 | 3.666 | 3.181 | 0.485 | 3.690 | 3.159 | 0.531 | 3.696 | 3.148 | 0.548 |
| LG4 | 3.781 | 3.107 | 0.674 | 3.797 | 3.061 | 0.736 | 3.818 | 3.043 | 0.775 |
| LS | 3.726 | 3.129 | 0.597 | 3.759 | 3.089 | 0.67 | 3.772 | 3.082 | 0.69 |

3.5.2. Electrochemical Impedance Spectroscopy

Figure 10a shows the Ac impedance diagram of the assembled lithium batteries LG3, LG4 and LS. It can be seen from the Figure 10a that the AC impedance diagrams of the three lithium batteries are composed of a semicircle in the high-frequency region and a straight line in the low-frequency region. The semicircle diameter of the high-frequency region is the resistance of charge transfer ($R_{CT}$), which reflects the interface resistance between the positive electrode material and the electrolyte in the lithium battery. The straight line in the low-frequency region is mainly related to Warburg impedance, which can reflect the diffusion ability of lithium ions in iron phosphate/lithium iron phosphate materials [30–32]. Figure 10b shows the relationship between Z′ and $\omega^{-1/2}/S^{-1/2}$ at low frequencies of the assembled lithium batteries. In this figure, the slope of the straight line represents the Warburg coefficient.

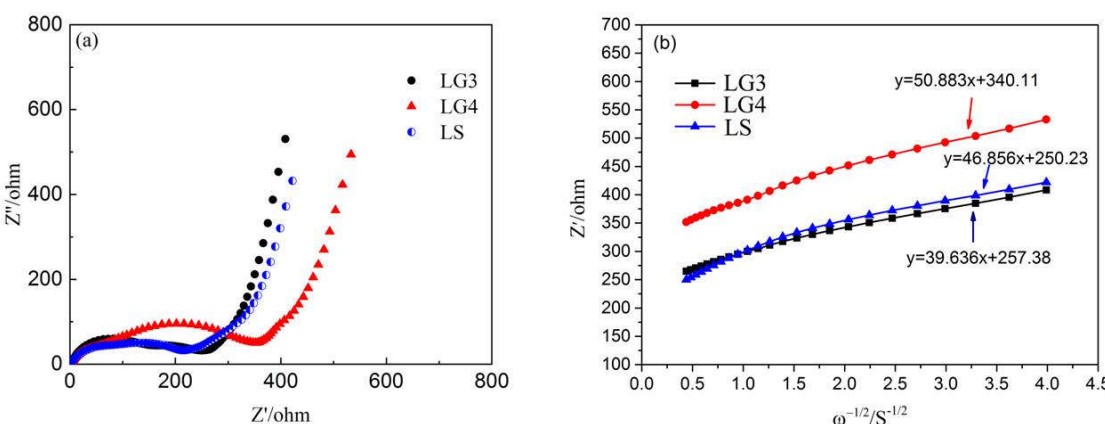

**Figure 10.** (**a**) The Ac impedance diagram of the assembled lithium batteries; (**b**) relationship between Z′ and $\omega^{-1/2}/S^{-1/2}$ at low frequencies of the assembled lithium batteries.

The lithium ions diffusion coefficient $D_{Li}$ is calculated from Equation (1):

$$D_{Li} = R^2T^2/\left(2\sigma^2A^2n^4F^4C^2\right) \tag{1}$$

where R is the gas constant, R = 8.3138462 ≈ 8.314 J/(mol·K); T is the absolute temperature, T = 298.15 K; σ is the slope of the straight line in the low-frequency region; A is the surface area of the battery electrode (cm²); F is Faraday's constant, F = 96485.3; ω is the AC angular frequency; S is the cross-sectional area between the electrolyte and the electrodes; C is the lithium ion concentration, which can be calculated from Equation (2):

$$C = \frac{n}{V} = \frac{m/M}{V} = \frac{\rho V/M}{V} = \frac{\rho}{M} \tag{2}$$

Table 5 shows the charge transfer resistance (Rct), Warburg factor (σ) and lithium ions diffusion coefficient ($D_{Li}$) of the assembled lithium batteries. It can be seen from Table 5 that the lithium ion diffusion coefficients of lithium batteries LG3 and LG4 are greater than that of lithium battery LS. This shows that the transmission performance of lithium ions in LG3 and LG4 lithium batteries is better, which is conducive to charging and discharging under high current densities. This may be because the battery separator used by lithium battery LS is coated with too many S-1 polymer particle adhesives, which affects the transmission of lithium ions.

**Table 5.** Charge transfer resistance (Rct), Warburg factor (σ) and lithium ions diffusion coefficient ($D_{Li}$) of the assembled lithium batteries.

|      | σ     | $R_{CT}$ (Ω) | $D_{Li}$              |
|------|-------|--------------|-----------------------|
| LG3  | 39.63 | 125          | $1.40158 \times 10^{-13}$ |
| LG4  | 50.88 | 323.2        | $1.00292 \times 10^{-13}$ |
| LS   | 46.85 | 183.6        | $8.50456 \times 10^{-14}$ |

### 3.5.3. Charge and Discharge Performance Test

Figure 11 shows the cyclic performance curves of assembled lithium batteries at different discharge rates. The data relating to discharge capacity and capacity retention rates of assembled lithium batteries at different discharge rates are shown in Table 6. As can be seen from Figure 11 and Table 6, the discharge capacity of the assembled lithium batteries gradually decreases when the discharge rates increases. The discharge capacities of the assembled lithium batteries (LG3, LG4 and LS) are higher than those of the assembled lithium batteries prepared by the separators coated without polymer particles adhesives (Uncoated) when the discharge rate is 0.2 C or 0.5 C. Furthermore, there is little difference in discharge capacity of the assembled lithium batteries when the discharge rate is 1 C. The reason for this phenomenon could be that the polymer particle adhesives on the separator are conducive to close contact with the material (LiFePO$_4$) on the positive electrode sheet, which can shorten the transmission path of lithium ions. In addition, the polar functional groups on the polymer particle adhesives can increase the wettability of the separator, making the separator and electrolyte wetting better, so that the lithium ions move rapidly during the charging and discharging process. However, when the discharge rate is 2 C, the discharge capacity of the lithium battery LS is 107.2 mAhg$^{-1}$, which is significantly lower than that of the other three lithium batteries. This is because there are too many polymer particle adhesives (S-1) coated on the separator in the lithium battery LS, which blocks the micropores on the separator and affects the shuttle of lithium ions.

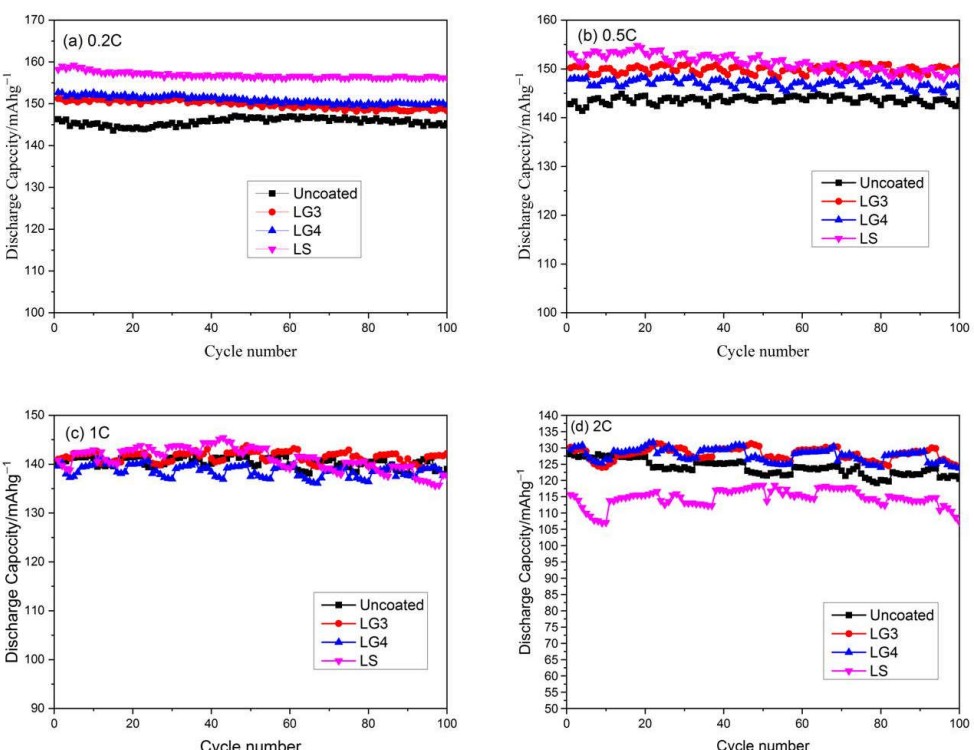

**Figure 11.** The cyclic performance curves of assembled lithium batteries at different discharge rates.

**Table 6.** The data of discharge capacity and capacity retention rate of assembled lithium batteries at different discharge rates.

| | * Discharge Capacity/mAhg$^{-1}$ | | | | * Capacity Retention Rate/% | | | |
|---|---|---|---|---|---|---|---|---|
| | 0.2 C | 0.5 C | 1 C | 2 C | 0.2 C | 0.5 C | 1 C | 2 C |
| Uncoated | 145.3 | 143.6 | 139 | 120.5 | 99.2 | 100.53 | 91.07 | 98.96 |
| LG3 | 145.3 | 150.5 | 142.2 | 124.3 | 99.2 | 99.44 | 100.2 | 99.2 |
| LG4 | 145.3 | 146.2 | 137.7 | 123.8 | 99.2 | 99.49 | 99.36 | 98.6 |
| LS | 145.3 | 149 | 137.6 | 107.2 | 99.2 | 98.7 | 98.5 | 96 |

* After 100 cycles of charge and discharge.

Figure 12 shows the capacity retention rate of assembled lithium batteries at different discharge rates. As can be seen from Figure 12 and Table 6, the capacity retention rate of the assembled lithium battery LG3 at different discharge rates is more than 99%, which is greater than that of the other three lithium batteries. In addition, the capacity retention rates of the assembled lithium batteries LG4, LS and Uncoated are basically the same. This shows that the coating of polymer particle adhesives (G3, G4 or S-1) on the battery separator does not affect the capacity retention rate of the assembled lithium batteries.

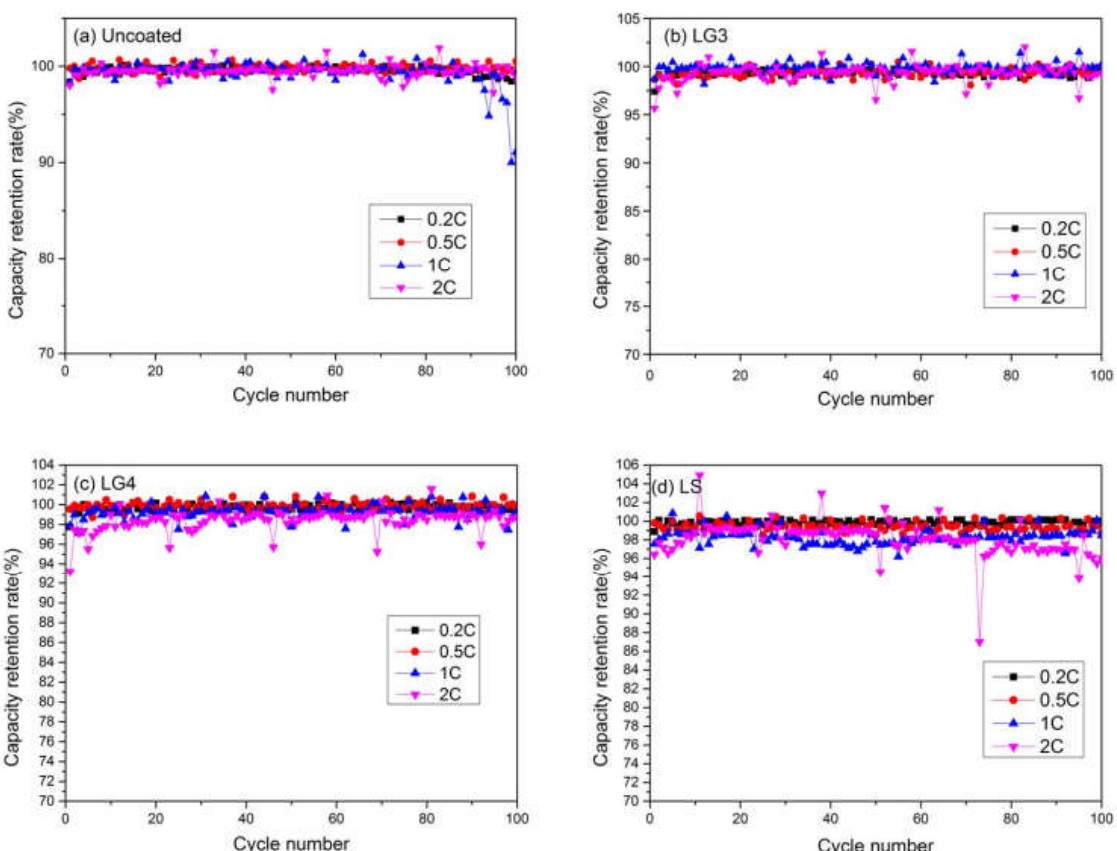

**Figure 12.** The capacity retention rate of assembled lithium batteries at different discharge rates. (**a**) shows the capacity retention rate of lithium battery at different discharge rates when the lithium battery separator was not coated without any polymer particle adhesives; (**b**) shows the capacity retention rate of lithium battery LG3 at different discharge rates when the lithium battery separator was coated with polymer particle adhesive G3; (**c**) shows the capacity retention rate of lithium battery LG4 at different discharge rates when the lithium battery separator was coated with polymer particle adhesive G4; (**d**) shows the capacity retention rate of lithium battery LS at different discharge rates when the lithium battery separator was coated with polymer particle adhesive S-1.

It is indicated that the polymer particle adhesives on the separator act as a binder, especially G3, which can enhance the bonding force between the positive electrode, the separator and the negative electrode. It also can prevent cracks, or even breakage, of the battery pole pieces, and ensure that the lithium ions can efficiently shuttle between the positive and negative electrodes of the assembled lithium batteries.

Therefore, it can be seen from the above electrochemical performance test and analysis results that coating a certain amount of polymer particle adhesive on the battery separator does not affect the performance of the lithium battery. In addition, due to the existence of the polymer particle adhesive, the adhesion between the battery separator and pole piece is enhanced, which is conducive to the stability and safety of lithium batteries, and can improve the preparation efficiency of lithium batteries.

## 4. Conclusions

Four kinds of polymer particles were synthesized and used as adhesives in the separators of lithium batteries. The polymer particle adhesives G3 and G4 are spherical particles with uniform size when the mass ratio of ethanol to water is 85:15. When polymer particle adhesive G3 is coated on the battery separator, the adhesion value between battery separator and pole piece is higher than that of polymer particle adhesives G4 and S-1, which is due to the low glass transition temperature of G3 (Tg = 33 °C).

The CV diagrams analysis shows that the assembled lithium battery LG3 has the smallest polarization and the best reversible performance during the charging and discharging process. It can be seen from the simulation calculation in the AC impedance diagram that the $D_{Li}$ value of the assembled lithium battery LG3 is $1.40158 \times 10^{-13}$, which is the maximum value. The results of CV and AC impedance diagrams of the assembled lithium battery LG3 indicate that the polymer particle adhesive G3 can not affect the redox peak of electrode materials and the transmission performance of lithium ions.

It can be seen from the test of charge and discharge performance that the capacity retention rate of the assembled lithium battery LG3 at different discharge rates is more than 99%. It is also indicated that the polymer particle adhesive G3 can not only provide a strong adhesion between the battery separator and pole piece, but also can not affect the discharge capacity and capacity retention rate of assembled lithium batteries at different discharge rates. The adhesion between the battery separator and pole piece is enhanced, which is conducive to the stability and safety of lithium batteries, and can improve the preparation efficiency of lithium batteries.

**Author Contributions:** Conceptualization, G.H. and H.W.; methodology, G.C.; formal analysis, Z.L.; supervision, H.H.; funding acquisition, S.G. All authors have read and agreed to the published version of the manuscript.

**Funding:** This work was supported in part by the Post-Doctoral Foundation Project of Shenzhen Polytechnic 6021330017K0, in part by the National Natural Science Foundation of China (Grant No. U2133213, 52071332), in part by the Department of Science and Technology of Guangdong Province (Grant No. 2019QN01H430, 2019TQ05Z654), in part by the Science and Technology Innovation Commission of Shenzhen (Grant No. JCYJ 20180507182239617, ZDSYS20190902093209795).

**Institutional Review Board Statement:** Not applicable.

**Informed Consent Statement:** Not applicable.

**Data Availability Statement:** Data is contained within the article.

**Conflicts of Interest:** The authors declare no conflict of interest.

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
