# Peer review of "Application of a New Polymer Particle Adhesive for Lithium Battery Separators"

_coatings, doi:10.3390/coatings13010021_

Round 1
Reviewer 1 Report
The authors synthesized four kinds of polymer
particles and used as adhesives separators
in the lithium battery. The electrochemical performance test of
the assembled lithium battery shows that the polymer particle adhesive G3 can
not only provide a strong adhesion between the battery separator and pole
piece, but also can not affect the discharge capacity and capacity retention
rate of assembled lithium batteries at different discharge rates and is
conducive to the stability and safety of lithium batteries and can improve the
preparation efficiency of lithium batteries. I think the ms can be considered
for publication, but after tackling out the following serious comments:
1. It is good to study and compare performance of four polymers, but where is the benchmark or commercial material, for example one of the polyolefin separators. Especially you conveyed this very well in the introduction. If you consider LS as a benchmark, please write more details, or clarify throughout the ms.
2. I would like to see DSC results for determination Tg and would be better to use the DSC curves in the paper to see the thermal behavior of your polymers.
3. In Figure 10, the colors in a and b are not matched. Please revise all figures in the ms. At the same time, the authors have to define ω and S in Figure 10b.
4. The authors showed include charge-discharge curves in the paper to see their cells charge and discharge behavior. I wonder why the authors excluded the original data in many parts throughout the manuscript.
5. The conclusion need to be revised, the authors should write a solid and clear summary about their work!

Reviewer 2 Report
This article reports an interesting work on the development of Lithium Battery Separator
Authors must describe in detail the innovation of this work compared to the literature and also the advantage of this material over others.
More details about sample preparation are needed and the results should be further discussed.
Round 2
Reviewer 1 Report
LS should be defined in the manuscript
Reviewer 2 Report
All comments have been included in the manuscript.
